# Detection of Viable but Non-Culturable (VBNC)-*Campylobacter* in the Environment of Broiler Farms: Innovative Insights Delivered by Propidium Monoazide (PMA)-v-qPCR Analysis

**DOI:** 10.3390/microorganisms11102492

**Published:** 2023-10-04

**Authors:** Benjamin Reichelt, Vanessa Szott, Kerstin Stingl, Uwe Roesler, Anika Friese

**Affiliations:** 1Institute for Animal Hygiene and Environmental Health, Freie Universität Berlin, 14163 Berlin, Germany; benjamin.reichelt@fu-berlin.de (B.R.);; 2Institute of Food Safety and Food Hygiene, Freie Universität Berlin, 14163 Berlin, Germany; vanessa.szott@fu-berlin.de; 3Department of Biological Safety, German Federal Institute for Risk Assessment (BfR), 10589 Berlin, Germany

**Keywords:** *Campylobacter*, environment, viable but non-culturable (VBNC), persistence, broiler, viability (v)-qPCR, PMA

## Abstract

Campylobacteriosis cases in humans are of global concern, with high prevalence rates in the poultry reservoir considered the most important source of infection. Research findings show *Campylobacters*’ ability to enter a viable but non-culturable (VBNC) state, remaining “viable” but unable to grow on culture media. We explored the persistence of VBNC states in specific environments, particularly at broiler farms, as this state may lead to an underestimation of the present *Campylobacter* prevalence. For VBNC detection, a propidium monoazide PMA-dye viability qPCR (v-qPCR) was used in combination with cultivation methods. We examined samples collected from broiler farm barns and their surroundings, as well as chicken manure from experimental pens. In addition, the tenacity of culturable and VBNC-*Campylobacter* was studied in vitro in soil and water. In a total of three visits, *Campylobacter* was not detected either culturally or by v-qPCR (no *Campylobacter* DNA) in the environment of the broiler farms. In four visits, however, VBNC-*Campylobacter* were detected both inside and outside the barns. The overall prevalence in environmental samples was 15.9% for VBNC-*Campylobacter*, 62.2% for *Campylobacter* DNA, and 1.2% for culturable *C. jejuni*. In the experimental pens, no cultivable *C. jejuni* was detected in chicken manure after 24 h. Strikingly, “VBNC-*Campylobacter*” persisted even after 72 h. “VBNC-*Campylobacter*” were confirmed in barn surroundings and naturally contaminated chicken manure. Laboratory studies revealed that VBNC-*Campylobacter* can remain intact in soil for up to 28 days and in water for at least 63 days, depending on environmental conditions.

## 1. Introduction

Poultry meat is a significant source of *Campylobacter* infections, with 127,840 recorded cases in the EU in 2021 [1]. The colonization of poultry at the farm level plays a crucial role in how *Campylobacter* enters the food chain. Most cases (20–30%) of human campylobacteriosis in the EU are attributed to the consumption of poultry meat. A significant proportion (50–80%) is thereby causally linked to the high prevalence of *Campylobacter* (*C*) *C. jejuni* and *C. coli* in the poultry reservoir. *Campylobacter* occurrence in poultry production is currently associated with emerging antimicrobial resistances to antibiotics. [2,3,4,5,6]. Despite extensive research at broiler farms, the knowledge to understand in which ways *Campylobacter* manages to colonize new flocks despite proper interventions remains incomplete. Some studies have provided valuable insights into the epidemiological situation using cultivation and molecular epidemiology. Notably, the environment is frequently mentioned as a reservoir for *Campylobacter* at chicken farms. However, the results of different studies suggest that cultivating *Campylobacter* in broiler farm environments remains challenging [4,7,8,9,10]. This could be attributed to the limited persistence of *Campylobacter* when exposed to various environmental conditions. Exposure to various environmental stressors, including oxidative stress, starvation, osmotic stress, temperature, pH, and UV light, has been discussed to induce a viable but non-culturable (VBNC) state in *Campylobacter* [11,12,13,14,15]. The VBNC state of *Campylobacter* was first described by Rollins and Colwell for its survival in natural aquatic environments [16]. Subsequent studies of food safety and primary production examined conditions within the poultry processing chain, with a particular focus on chicken carcasses, meat rinses, and raw milk [17,18,19,20]. The persistence of *Campylobacter* in the environment of broiler farms and subsequent colonization of broilers may also be related to the VBNC state. Previous studies have demonstrated that VBNC-*Campylobacter* (*C. jejuni*) can resuscitate in vivo or under laboratory conditions, as well as express pathogenicity [11,20,21]. Furthermore, recent research has underscored the importance of VBNC *Campylobacter* in food processing conditions. [22]. As culture-based methods cannot detect VBNC-*Campylobacter*, polymerase chain reaction (PCR) methods are used; however, they only amplify the total DNA (from both viable and dead cells). PMA pre-treatment combined with qPCR has been used in various investigations to confirm viable *Campylobacter* [17,19,23,24,25]. However, PMA treatment may fail to fully inactivate the remaining signal of dead cells in qPCR [19]. To address this, an internal sample process control (ISPC) was developed, which monitors dead cell signal reduction and DNA loss during extraction that allows accurate quantification [26,27]. In the current study, this sophisticated PMA dye-supported viability (v)-qPCR approach that was recently validated for meat rinses in line with ISO 16140-2:2016 [26] was used to examine various environmental matrices. The investigation focused on a one-year sampling campaign aimed at sampling the environment of three broiler farms in Germany. Additionally, naturally contaminated chicken manure obtained from experimental pens was analyzed. To detect and quantify VBNC-*Campylobacter* in poultry and environmental samples, a pretreatment step involving PMA and ISPC was employed. Subsequently, following the photoactivation of the dye, the samples were analyzed using qPCR. To further expand the understanding of VBNC states in the environment, (i) the tenacity of culturable *Campylobacter*, (ii) the stability of VBNC *Campylobacter* induced in raw milk, and (iii) the transition of culturable *C. jejuni* into the VBNC state in vitro was investigated.

## 2. Materials and Methods

### 2.1. Study Design

For the first part of the study (field trial), seven visits to broiler farms (A-C) were conducted between November 2019 and September 2020 (Table 1). All farms followed an all-in/all-out system and provided broilers ad libitum access to feed and water via drinker nipples with trays. Ross 308 broilers (farm A and C) were reared at a stocking density of 39 kg/m^2^ for 36–42 days, while Hubbard broilers (farm B) were stocked at 25 kg/m^2^ for 60 days. The chickens received a three-phase feeding diet matching the commercial standards. Access to outdoor areas was not provided. Thinning procedure was carried out approximately one week before the entire flocks were removed. The three rural farms are surrounded by fields, forests, and small artificial waterways with adjacent lakes present at distances of 0.5 to 1.5 km. The farms followed several biosecurity measures, including personal hygiene practices, disinfectant footbaths or mats and cleaning and disinfection of the broiler houses as specified in guidelines by the German Agricultural Society (DLG) and the German Veterinary Society (DVG).

The second part of the study (experimental trial 1) was conducted at the experimental facilities of the Centre for Infection Medicine within the Department of Veterinary Medicine at Freie Universität Berlin. The investigation was carried out after the removal of the flocks. The natural *Campylobacter*-contaminated chicken manure (harboring the strain BfR-CA-14430) [28] from four separate animal rooms was investigated over a period of 72 h. Chicken manure was stored under stable environmental conditions: a temperature of approximately 20 °C, a relative humidity (RH) of about ~50%, an air exchange rate of 15 times per hour in the room, and artificial daylight of 400 lux.

The third part of the study was a laboratory-based in vitro study (experimental trial 2), which was split into three different trials. The first trial aimed at investigating the tenacity of cultivable *Campylobacter*. In the second trial, the stability of VBNC *C. jejuni* induced in raw milk was observed and determined. In the last part, a combination of both trials and a further determination of the transition of cultivable *C. jejuni* into their VBNC state was investigated. Soil and water were used as experimental matrices, and all trials were carried out in three different microhabitats, each characterized by unique features as outlined in Table 2. The soil originating from the Berlin region displays a light texture, characterized by a substantial presence (over 80%) of sand particles, a minor content (less than 10%) of clay particles, a moderate amount (10–40%) of silt particles, and a neutral pH 7. The water (drinking water with drinking water quality, pH 7.0) used for the experiment was obtained from the drinking water system of the experimental animal husbandry. The matrices were stored in sterile 120 mL specimen containers (VWR, Radnor, Pennsylvania).

### 2.2. Sampling and Pre-Treatment

#### 2.2.1. Field Trial

Sampling was conducted at each farm after the broiler thinning procedure was carried out. Environmental samples (air, boot swabs, gauze swabs, and water) were collected outside the barns, as described previously [4]. After sample collection, the samples were transported in a cooling box (~4 °C) to the laboratory and analyzed within 2 h. One pair of boot swabs were homogenized in filtered blender bags by shaking for 120 s in 100 mL of peptone water (PW) using the “fast” (120 rounds per minute (rpm)) program of a laboratory stomacher. Gauze swabs were similarly homogenized in 50 mL PW. Chicken manure was initially homogenized in a 120 mL specimen tank using a sterile spatula. Subsequently, 5 g of the homogenized sample was diluted at a ratio of 1:10 in PW and further homogenized as described above using the stomacher. Water was homogenized by gently vortexing. After its initial homogenization, 50 mL of each pretreated sample was filtered through folded filters 5–13 µm (Rotilabo^®^-Faltenfilter, Typ 601P Carl Roth, Karlsruhe, Germany) and sterile glass funnels. Filtered samples were centrifuged at 4 °C (11,000× *g*) for 15 min, the supernatant discarded, and the pellet resuspended in 3 mL PW. The final sample was then divided into aliquots as follows: 1 mL reserved for cultivation and 2 mL for differentiation of live and dead cells using quantitative polymerase chain reaction (v-qPCR). Air samples (total volume of 1000 L/1 m^3^) were collected as described previously [4] using Coriolis^®^ µ cones were transferred to a 15 mL sterile screw cap tube (Sarstedt, Nümbrecht, Germany), and the sample was then centrifuged at 4 °C (11,000× *g*) for 15 min. The supernatant was discarded, and the pellet was resuspended in 3 mL PW and aliquoted. From each water sample, 50 mL was either filtered and centrifuged or directly centrifuged, depending on the level of visible pollution.

#### 2.2.2. Experimental Trial 1

Chicken manure was collected at 24 h intervals from 0 to 72 h after flock removal from four separate animal rooms. Per room, chicken manure was divided into four different areas (four biological replicates), which were distinguished in terms of moisture and dryness. Approximately 30–50 g of manure was collected with sterile spatulas and transferred into sterile 120 mL specimen containers. Manure samples were then prepared and handled as described above.

#### 2.2.3. Experimental Trial 2

Water from the experimental trial 2 was vortexed, and 3 mL of the homogenized mixture was transferred to a sterile 15 mL tube, gently vortexed again, and aliquoted as described previously. Soil samples were collected using sterile spatulas and placed into sterile screw cap tubes (50 mL) (Sarstedt, Nümbrecht, Germany). Afterwards, samples were diluted 1:2 in 3 mL of PW and homogenized by vortexing. For the determination of colony-forming units (CFU) (cultivation), 1 mL of the suspension was used. For PMA-v-qPCR analysis, 2 mL of the sample mixture was centrifuged at 600× *g* for 1 min to eliminate soil particles, sand grains, and other organic matter that could not be readily filtered and may have a detrimental impact on DNA extraction and v-qPCR. Subsequently, the supernatant was transferred to a sterile tube and centrifuged again at 600× *g* for 1 min. Finally, the supernatant was homogenized and aliquoted as described above.

### 2.3. Inoculation Strain and Growth Conditions

For the laboratory-based study in experimental trial 2, the *C. jejuni* strain (BfR-CA-14430), preserved at −80 °C, was cultivated on Columbia agar supplemented with 5% sheep blood (ColBA, Oxoid, Thermo Fisher Scientific Inc., Waltham, MA, USA) for 24 h at 42 °C under microaerobic conditions (5% O_2_, 10% CO_2_ and 85% N_2_) in a tri-gas incubator (CB 160; Binder, Germany). After sub-culturing in BHI with twice the amount of Growth Supplement (SR0232; Oxoid, Wesel, Germany) for 18 ± 2 h, cells were suspended in BHI and adjusted to an optical density of 0.2 at a wavelength of 600 nm (OD_600_), equivalent to approximately 9 log_10_ cell counts per ml as reported earlier [25]. This suspension was spiked into soil or water to achieve a final concentration of ~8 log_10_ CFU/g or ml. The VBNC-*C. jejuni* cells (BfR-CA-14430) in raw milk (~7 log_10_ viable cells/mL) were provided by the German National Reference Laboratory (NRL) for *Campylobacter* [20] and spiked in soil and water at a final concentration of ~5 log_10_/g or ml.

### 2.4. Cultivation Methods for the Studies

Quantification was evaluated via colony forming units (CFU) following ISO 10272-2:2017 [29]. Pre-treated samples were diluted 10-fold in BHI and plated in duplicate on modified cefoperazone deoxycholate agar (mCCDA) (Oxoid, Thermo Fisher Scientific, Waltham, MA, USA). Samples that contained low levels of *Campylobacter* in experimental trial 1 were subjected to quantitative assessment using enrichment, following the ISO 10272-1:2017 [30] procedure B. In brief, samples were diluted at a 1:10 ratio in Preston Broth, (PB) supplemented with Preston *Campylobacter* selective Supplement (SR0117; Oxoid, Wesel, Germany), Growth Supplement (SR0232; Oxoid, Wesel, Germany), and defibrinated horse blood (SR0050; Oxoid, Wesel, Germany), then incubated in a microaerobic atmosphere at 41.5 °C for 24 h. Enriched cultures were inoculated onto selective mCCDA plates using a sterile 10-μL loop, followed by incubation at 41.5 °C under microaerobic conditions for another 48 ± 2 h. Putative colonies were isolated and streaked on Columbia blood agar with 5% sheep blood and then incubated as described above. Colonies were analyzed using matrix-assisted laser desorption ionization time-of-flight mass spectrometry (MALDI-TOF MS, Bruker Microflex^®^ system). When *Campylobacter* growth in experimental trial 2 was not achievable following ISO 10272-2:2017, putative VBNC cell suspensions (100 µL) from 1 mL reserved suspension from pre-treatment were transferred to supplemented PB and incubated under similar conditions for up to 72 h to confirm the absence of viable *C. jejuni*.

### 2.5. Determination of VBNC Campylobacter with qPCR

The method followed a previously established protocol [26,27]. In brief, for each pretreated sample, two “working” samples of each 1 mL were prepared. One sample was stained with 2.5 µL of a 50 μM PMA solution (viable cells) while the other sample served as a control to monitor total DNA of all cells (viable and dead cells). To monitor the reduction of the dead cell signal by PMA, an internal sample process control (ISPC) at high concentration was included in the PMA-treated samples [27]. The mixture was briefly vortexed and incubated in a laboratory thermomixer at 700 rpm for 15 min at 30 °C without light exposure (darkened room). Following incubation, samples were cross-linked for 15 min using a PMA-Lite™ LED Photolysis Device (Biotium Inc., Landing Parkway, Fremont, CA, USA). Subsequently, ISPC at low concentration was added to both PMA-treated and untreated samples, the samples were gently vortexed and centrifuged for 5 min at 16,000× *g* at 4 °C. The latter addition of low concentration of ISPC guarantees that putative DNA losses during extraction are additionally detected in individual samples [26]. The supernatant was discarded, and cell pellets were stored until DNA extraction at −20 °C. Genomic DNA was extracted using the GeneJet Genomic DNA Purification Kit (Thermo Fisher Scientific Inc., Waltham, MA, USA), according to the manufacturer’s instructions.

The targets of the qPCR assay have been previously described (Table 3). Each qPCR run employed genomic standards (*C. jejuni* NCTC 11,168 or *C. sputorum* DSM 5363 (ISPC). Standards were used in duplicates in a set of serial dilutions (5000, 500, 50, 20, and 10 genomic copies) per reaction to generate standard curves for quantification. The National Reference Laboratory for *Campylobacter* provided the genomic standards in dried stabilized DNA aliquots as described previously [27].

The triplex v-qPCR method was employed using the fluorophore combination Jos-P-FAM, Csput-P-Cy5 and IPC-ntb2-P-HEX [26]. The triplex-master mix was prepared as described by the standard operating procedure (SOP) (Suppl. Information 1–2) [26]: 1× Platinum Taq buffer, 2.5 mM MgCl2, 0.2 mM of each dNTP (Thermo Fischer Scientific, USA), 0.06 × ROX (Life Technologies, USA), 500 nM of each Jos-F1 and Jos-R1 primer, 500 nM of each Csput-F and Csput-R primer, 300 nM of each IPC-ntb2-F and IPC-ntb2-R primer, and 100 nM of each dark quenched (IPC-ntb2-P-HEX, Jos-P-FAM, and -Csput-P-Cy5) (Biomers GmbH, Ulm, Germany) (refer to Table 3) and 2U Platinum^TM^ Taq DNA Polymerase (Invitrogen, Thermo Fisher Scientific Inc.). The v-qPCR program started with a 15 min incubation at 95 °C, followed by 45 cycles of 30 s at 95 °C and 1 min at 60 °C (measure fluorescence) and 30 s at 72 °C for Platinum^TM^ Taq DNA Polymerase.

In cases where amplification was putatively hindered by elevated levels of humic acid, the PerfeCTa^®^ qPCR ToughMix^®^ (Quantabio, Beverly, MA, USA) was employed as recommended. The triplex qPCR method is effective for most samples, but in cases where *Campylobacter* spp. exceed the maximum limit of quantification (4.7 log_10_ genome equivalents per ml), it can hinder the ISPC signal and render it unsuitable for quantitative analysis [26]. In those cases, two duplex qPCR assays were used instead [27].

### 2.6. Statistical Analysis

All quantitative data were compiled in a Microsoft Excel spreadsheet. Statistical data analysis and graphs were created using GraphPad Prism 9.1.0 (221) (2020, GraphPad Software, 2365 Northside Dr. Suite 560, San Diego, CA 92108, USA). Mann-Whitney *U*-test (two-tailed) and Kruskal–Wallis test with Dunn’s multiple comparison test were employed. Differences were statistically different when *p* < 0.05 (**** *p* < 0.0001).Values are calculated as the mean (M) with a standard deviation (SD).

## 3. Results

### 3.1. Field Trial

In total, seven sampling time points were analyzed for the presence of *Campylobacter* spp. Of these, *Campylobacter* spp. was detected only at four time points (1, 2, 6 and 7), while at visits 3–5, all samples (*n* = 12) from the barns tested negative for *Campylobacter* independent of the applied detection methods. Furthermore, no cultivable *C. jejuni* (CFU) or total *C. jejuni* DNA (e.g., from dead cells) was found in the environment (*n* = 72) at visits 3–5. Overall, 15.9% (13/86) of the environmental samples, were confirmed to be positive for viable *C. jejuni* cells when the barns were positive for *C. jejuni*, which corresponded to log_10_ viable *C. jejuni* (Cj)/sample after PMA treatment. Cultivable *C. jejuni* were confirmed in only one out of 86 environmental samples (water) (1.2%). Viable *C. jejuni* was not detected in any of the air samples (*n* = 28). However, when air samples were tested for *C. jejuni* DNA without PMA, all 28 samples (100%) were positive with a concentration of 2.5 ± 1 log_10_ dead Cj/m^3^ (*p* <0.0001). In boot swabs from the environment (*n* = 24), viable *C. jejuni* was determined in 11 samples (45.8%) at a concentration of 3.2 ± 0.8 log_10_ viable Cj/boot swab sample (*p* <0.0001). In environmental gauze swabs, *C. jejuni* DNA from dead cells was identified in 10 samples (*n* = 18), with a concentration of 2.5 ± 0.2 log_10_ dead Cj/gauze swab sample (*p* < 0.0001).

Using PMA-dye, viable *C. jejuni* was identified in one gauze swab at a concentration of log 1.5 log_10_ viable Cj/gauze swab. However, this result, along with a positive finding in one water sample of 1.8 log_10_ viable Cj/water sample (*n* = 16), was below the validated limit of quantification (LOQ of 2 log_10_) of the method and, thus, interpreted as semi-quantitative values. Nevertheless, equal quantities were cultivated in the water sample (1.5 log_10_ CFU/water sample). Regarding barn matrices, (Figure 1B) chicken manure samples from the inside of the barns revealed high levels of cultivable *C. jejuni*, with 4.5 ± 1.6 log_10_ CFU/mL. Moreover, v-qPCR determined 3.3 ± 0.3 log_10_ viable Cj/mL. This showed a high correlation between viable cells and CFU counts (no significant difference (*p* = 0.44), providing an accurate estimation by PMA despite potential negative matrix effects. Strikingly, *Campylobacter* was not cultivated from one chicken manure sample while simultaneously v-qPCR determined 3.5 log_10_ viable Cj/5 g manure. The corresponding boot swab sample, probably soiled with fresh fecal or cecal droppings, contained ~6.5 log_10_ viable *C. jejuni* with v-qPCR and CFU per boot swab (refer to Figure 1B).

### 3.2. Experimental Trial 1

Based on the results of the field trial, an experimental investigation was conducted to further analyze *C. jejuni* persistence and possible transition to the VBNC state in naturally contaminated chicken manure. In this study, manure samples (four biological replicates) were collected after removal of the flocks from separate experimental animal rooms and analyzed using CFU and v-qPCR. The CFU method yielded 0 CFU/g (*n* = 16) of *C. jejuni* in all examined manure samples immediately after the removal of the flocks (sampling point 0). In contrast, the qualitative detection of *C. jejuni* using enrichment culture yielded positive results in the samples. Cultivable *C. jejuni* was detected for up to 24 h by qualitative detection methods. Interestingly, v-qPCR revealed that viable *C. jejuni* was quantifiable in most of the pre-treated samples at time point 0, with concentrations of 3.7 log_10_ ± 0.8 viable Cj/5 g manure (*n* = 16) (Figure 2). Throughout the 0–48 h investigation, viable counts remained relatively constant (Figure 2). After 72 h 3.2 ± 0.9 log_10_ viable Cj/5 g manure (*n* = 9) were still determined.

### 3.3. Experimental Trial 2

In the first trial, the tenacity of cultivable *Campylobacter* was determined in soil in open and closed containers in the laboratory at room temperature (RT) (A), in an incubator at high humidity at RT (B) or in the refrigerator at 4 °C at ambient humidity (Table 2). During this trial, the *C. jejuni* strain BfR-CA-14430 rapidly lost its cultivability in open containers under laboratory conditions at RT (A), thus being additionally exposed to daylight (Figure 3). From an initial concentration of ~8 log_10_ CFU/g, only 3.1 log_10_ CFU/g were observed in one of the three replicates after 24 h. Subsequently, *Campylobacter* was only qualitatively detected after enrichment procedures. In contrast, higher concentrations of 7.8 ± 0.11 log_10_ CFU/g *C. jejuni* (*n* = 3) were observed using containers with a closed lid after 24 h. The inactivation rate in soil using a closed container in habitat A was consistent with the inactivation rate in the incubator setting (B), regardless of whether it was stored in an open or closed container (Figure 3). *C. jejuni* demonstrated a notably longer cultivation period and higher quantities under refrigerator conditions (C) (Figure 3). Using an open container, quantitative detection was achievable in habitat C for up to 7 days (Figure 4) (2.2 ± 0.07 log log_10_ CFU/g (*n* = 3)), and qualitative detection for up to 11 days. Using a closed container, cultivation capacity was further extended in habitat C, with 6.9 ± 0.16 log_10_ CFU/g (*n* = 3) detected on day 11. Afterward, cultivation capacity rapidly declined in habitat C, as well and quantitative cultivability was achieved for the last time on day 14 (see Figure 3).

In the second trial, the stability of laboratory-induced VBNC *Campylobacter* generated in raw milk was investigated. Viable but not culturable cells of *C. jejuni* BfR-CA-14430 were introduced into two different matrices (water and soil) at an approximate initial concentration of ~5 log_10_ viable Cj/g as determined by v-qPCR. Under condition A, viable *C. jejuni* were recovered from soil at a concentration of 3.46 ± 0.37 log_10_ viable Cj/g (*n* = 3) after 24 h. Within five days, viable counts decreased gradually by an additional ~1.5 log level to 2.17 ± 0.18 log_10_ viable Cj/g (*n* = 3) (Figure 4I). Similarly, the viable counts of *C. jejuni* in soil decreased rapidly in habitat B. By day four, the determined viable counts of *C. jejuni* decreased from the initial 3.75 ± 0.7 log_10_ viable Cj/g (*n* = 3) to 1.96 ± 0.12 log_10_ Cj/g (*n* = 3). Moreover, condition B extended the stability in soil to day 11 (1.82 ± 0.17 viable Cj/g). In contrast, under condition C, an initial recovery of viable cells was detected after 24 h of 3.86 ± 0.2 log_10_ viable Cj/g (*n* = 3), while an average of 2.7 ± 0,3 log_10_ viable Cj/g was detectable until the end of the trial (day 25) in soil. In water, the introduced 5 log_10_ viable cells were detectable for up to 25 days in conditions A and B. In habitat C, on the other hand, there was no relevant decrease in viable cells even during prolonged storage over 109 days, when no viable Cj could be retrieved under conditions A and B (see Figure 5).

In the third trial, both matrices (water and soil) were spiked with cultivable *C. jejuni* BfR-CA-14430 to achieve an initial concentration of approximately ~8 log_10_ CFU/g soil or ml water. Then, the investigation aimed to explore the loss of cultivability and the possible transition into VBNC. As observed in the tenacity study, the initial recovered amount (d0) was 7.0 ± 0.8 log_10_ CFU/g (*n* = 3) in soil. Under conditions A, CFU decreased rapidly after one day (d1) to 3.58 ± 0.23 log_10_ CFU/g when compared to the other microhabitats (B, C). Thereafter, *C. jejuni* could only be qualitatively detected after 48 h (d2) in habitat A in soil (Figure 5I). After day three (d3), cultivable *C. jejuni* were not detected using qualitative detection, but with v-qPCR, 3.2 ± 0.7 log_10_ viable Cj/g (*n* = 3). Equivalent amounts could be detected until the end of the experiment on day 21 (Figure 5I). In contrast, cultivable *Campylobacter* were qualitatively detectable until day 6 in habitat B, when 5.0 ± 0.44 log_10_ (*n* = 3) viable counts were detected by v-qPCR. Viable counts were continuously detected until day 21. Furthermore, CFUs were determined in soil until day 12 in habitat C. With the loss of cultivability after day 15, viable (3.97 ± 0.2) log_10_ viable Cj/g (*n* = 9) were determined) counts in habitat C with v-qPCR, which were stable until the end of the experiment (d28) (Figure 5I).

Consistent with observations from soil samples, a rapid decline in the cultivability of *C. jejuni* in water (habitat A) was observed. By day two (d2), only one of the three replicates showed a detectable level of 2.2 log_10_ CFU/mL of cultivable *C. jejuni*. Subsequently, qualitative detection was last possible in one replicate on day 3. However, the v-qPCR analysis (Figure 5) revealed a substantial presence of viable *C. jejuni* cells (7.9 ± 0.05 log_10_ viable Cj/g (*n* = 3). The gradual decay of viable cells by ~4 log_10_ was observable using v-qPCR until day 63 (Figure 5II).

In comparison, in habitat B, the loss of cultivability of *C. jejuni* was delayed by two days (Figure 5II). After the loss of cultivability, viable cells were successfully detected using v-qPCR (Figure 5II). At the end of the experiment (d62), viable cells at a concentration of 5.8 ± 1.4 log_10_ viable Cj/mL (*n* = 3) were still determined. In habitat C, the conditions preserved viable cells until day 63. Remarkably, under these conditions, *C. jejuni* remained cultivable for approximately three times longer compared to the other habitats, as shown before. Specifically, culturable *C. jejuni* were observed until day 14, and viable cells were detected until day 63 in water under condition C at cooling temperature (Figure 5II).

## 4. Discussion

### 4.1. VBNC Campylobacter in the Environment of Broiler Farms (Field Trial)

The one-year investigation of seven broiler farms revealed that, in total, 15.9% of the environmental samples from *Campylobacter*-positive broiler flocks contained viable cells, while only one sample (1.2%) could be retrieved as CFU. In contrast, the absence of *Campylobacter* DNA (from dead and viable cells) in the environment of broiler farms correlated with the absence of cultivable *Campylobacter* in the barns. *Campylobacter* was primarily absent during the winter. This seasonal phenomenon was previously described [32]. Although the detection of potential VBNC *Campylobacter* was infrequent, these recent data show VBNC *Campylobacter* presence which suggest *Campylobacter* transmission into the environment. Specifically, the determination of viable *C. jejuni* cells with v-qPCR in one water sample with simultaneous observation of lower CFU loads indicated a potential transition into the VBNC state. Therefore, it is reasonable to hypothesize that water bodies favor prolonged persistence while supporting the gradual transition of cultivable cells to the VBNC state. In support, reduced environmental stressors, such as low levels of dissolved oxygen and UV-light exposure in water, could be promoting longevity and thus a gradual transition of *Campylobacter* in the VBNC state [33,34]. Moreover, it has been suggested that the availability of organic matter (manure remnants) in water bodies provides ample nutrients that allow *Campylobacter* persistence [35]. Yet, it can be contended that nutritional and oxidative stress also promote rapid VBNC induction, as observed in a recent study [21]. In this study, most viable *C. jejuni* were primarily detected by v-qPCR after treatment with PMA in sock swab samples (45.8%), collected by walking a predetermined route at *Campylobacter*-positive farms. This phenomenon could potentially be elucidated by the presence of broiler manure residues in the surroundings, along with boot swabs frequently being contaminated with chicken manure. It is conceivable that contaminated manure is spread by personnel and their vehicles after partial depopulation, as this practice has been recently connected with *Campylobacter* transmission at broiler farms [36]. This assumption is also consistent with the results of the gauze swabs, as the sampled surfaces were barely in direct contact with contaminated chicken manure. Thus, a high proportion of *Campylobacter* DNA from dead cells was determined in gauze swabs, although one gauze swab was semi-quantitatively positive for viable *C. jejuni* with v-qPCR. This could be due to the increasing deposition and accumulation of ventilated broiler sheds and manure in the environment [37]. In contrast, VBNC *Campylobacter* was not determined in air samples (only *Campylobacter* DNA from dead cells was determined). These findings show the discrepancy between viable and dead cells, providing a new insight as previous research only utilized PCR methods without PMA and the ISPC [38,39]. Viable *C. jejuni* (potentially in the VBNC state) were observed more frequently during periods of ‘mild weather’ characterized by overcast, cloudy, and rainy conditions (Table 1). These environmental weather conditions could potentially trigger *Campylobacter* persistence and transition into the VBNC state. This hypothesis is supported by the observations obtained from laboratory survival experiments conducted in water and soil at different temperatures, in which water at cooling temperatures led to maximally enhanced survival of *C. jejuni* (independent of the culturable or VBNC state (Figure 5)).

### 4.2. VBNC Campylobacter in Naturally Contaminated Chicken Manure (Experimental Trial 1)

During the field trial, chicken manure and water were found to be two conceivable reservoirs of VBNC *Campylobacter* in the environment. The investigation of natural contaminated chicken manure from experimental pens aimed at the detection of VBNC *C. jejuni* in a partially controlled environment where both, the inoculation strain and the conditions of the experimental animal rooms were known. Overall, using cultural quantitative analysis, cultivable *C. jejuni* were not determined, although cultural qualitative detection was possible through enrichment directly after the removal of the flocks (0-h mark). This observation is consistent with previous research suggesting that *Campylobacter* may rapidly lose its cultivability in manure [40]. More importantly, a continuous number of viable cells was detected between the 24- and 48-h mark. However, it is important to highlight that the presence of turbidity resulting from total suspended solids (TSS) and organic components in manure had adverse impacts on the effectiveness of PMA inactivation in the used protocol, as previously mentioned [41,42]. Consequently, the successful inactivation of ISPC was achieved after pretreating the manure samples, a process that involved centrifugation and filtration. However, pretreatment might have contributed to a loss of total viable cells before addition of ISPC. Nevertheless, the application of pretreatment methods under experimental conditions resulted in the detection of viable *C. jejuni* counts, comparable to those obtained in the field trials of barns (~3.5 log_10_ viable Cj/5 g manure), indicating that any potential bias introduced by physical pretreatment of the samples was at least reproduced. These results indicate that VBNC *C. jejuni* remained stable in chicken manure for several days. Indeed, these observations confirm the hypothesis from the field trial, and substantiate the assumption that VBNC *Campylobacter* originating from manure can be released directly from the animal barn into the environment. Nevertheless, it is important to reiterate that the storage of the manure was carried out under controlled and stable conditions, within an experimental animal room. These conditions encompassed factors like: the absence of UV light exposure, variations in air exchange rate, and relative humidity, which could differ from the conditions found in the natural environment.

### 4.3. Persistence and Transformation of Campylobacter (Experimental Trial 2)

First, the natural decay and loss of cultivability of the *C. jejuni* strain in soil (placed in open (with desiccation) and closed containers (without desiccation)) were determined under laboratory conditions (A). In open containers, a rapid loss of *C. jejuni* cultivability (within one day) was observed at 21 °C and low RH, which was anticipated due to desiccation stress caused by low humidity [43]. In contrast, when placed in closed containers, *C. jejuni* was detectable by cultural detection methods for up to three days under laboratory (A) and incubator (B) conditions when using high relative humidity (99%). Desiccation appeared to be an important environmental driver for *C. jejuni* survival as its capacity was reduced by a loss of moisture, which might additionally lead to increased oxygen tension. Similarly, shifts in temperature were shown to correlate with *C. jejuni* viability. In particular, low temperatures around 4 °C were observed to extend the survival capacity of *C. jejuni* in soil. This observation is substantiated by an extensive body of research that has noted prolonged survival in both water and food matrices at 4 °C [20,44,45,46].

Second, the stability of VBNC *C. jejuni* BfR-CA-14430 (induced in raw milk) was investigated in the same microhabitats. Despite observing cell losses during the recovery process from the soil matrix, as discussed above, VBNC *C. jejuni* were still detected for an extended period, depending on the utilized microhabitat. However, it should be noted that soil components themselves, such as humic acid, may interfere with qPCR and may have a negative impact on qPCR [47]. These effects were ascertained using the ISPC, as this standard behaved similarly to the *C. jejuni* target. Hence, it was manageable to address this issue by implementing a more resilient DNA polymerase for the soil samples. This adjustment allowed us to accurately assess inactivation through PMA monitored by ISPC. Remarkably, within the laboratory environment (A), desiccation and oxygen stress had a less severe impact on the integrity of *C. jejuni* cells that were already in the VBNC state. Despite the lack of desiccation and moisture loss (induced by inoculation) in habitat (B), maintaining a relative humidity of 99%, the initial higher levels of viable cells observed in comparison to habitat (A) experienced a subsequent 2-log reduction by the end of the 11-day trial period. This decrease could be linked to other biotic factors within the soil matrix at the temperature range of 21–22 °C. At refrigerator conditions (C), however, viable *C. jejuni* cell counts remained stable for prolonged periods, again indicating that low temperatures are favorable for VBNC *C. jejuni*. Interestingly, storing milk-induced VBNC *C. jejuni* in water resulted in maintenance of viable *C. jejuni* counts during the experimental period of 25 days in habitats A and B and even 109 days in habitat C. This may imply that stress conditions in water have less impact on VBNC stability than those encountered in soil [33,48]. These findings are consistent with prior research that has examined VBNC *Campylobacter* in both water- and food-related matrices under comparable conditions [20,21,22,33].

Finally, the gradual transition of culturable *C. jejuni* into VBNC *C. jejuni* in soil and water was observed. Under laboratory conditions (A), rapid loss of cultivability in soil was observed as previously described. Thereafter, viable *C. jejuni* was detected using v-qPCR until day 28. Notably, the inoculation strain also lost its cultivability very rapidly in water (within three days) at RT. Overall, a ~4 log reduction of viable *C. jejuni* cells was observed in water samples over time at any condition but slower at refrigerating temperature. However, cells remained viable until the end of the investigation on day 63. It might be assumed that the exposure to daylight under condition A in open containers, filtered by glass windows, might have harmed viable cells over time due to the sensitivity of bacteria to photo-oxidative damage [34]. Under incubator conditions (B) of 21 °C, cultivable *C. jejuni* were detectable in both water and soil for up to five days by qualitative detection, suggesting that the matrix itself was negligible at high humidity. Moreover, viable *C. jejuni* cells were detectable until the end of the trials (in soil until day 28 and in water until day d63). These results are in line with prior observations of in milk-induced VBNC *C. jejuni*, which corroborates the aforementioned studies.

### 4.4. VBNC Campylobacter in Diverse Environmental Matrices

The results of the field study show that VBNC *Campylobacter* can persist in contaminated environmental matrices under favorable conditions. In the subsequent experimental trials, prolonged stability of viable *C. jejuni* in water was demonstrated by v-qPCR, especially at 4 °C. Therefore, it is of major interest to understand the potential risk of water-associated VBNC *C. jejuni* in the agricultural environment. Several studies assessed *C. jejuni* in different waters using qPCR [49,50,51,52]. However, these observations confirm and underline the hypothesis that microbiological enumeration considerably underestimates the fraction of viable *C. jejuni*. Thus, employing v-qPCR in conjunction with ISPC could offer additional insights into the presence of VBNC-*Campylobacter* in aquatic farm environments, as previously indicated [53]. The urgency for this is further supported by recent findings where the virulence of VBNC *C. jejuni* in primary chicken embryo cells (accordingly outgoing pathogenicity) was demonstrated [21]. As described earlier, at the farm level, VBNC *C. jejuni* may persist in manure residues or contaminations in the environment. Consequently, it is feasible that VBNC-*Campylobacter* are transmitted via contaminated manure that remains in the environment after flock removal and are introduced into subsequent flocks [10]. Furthermore, it was possible to detect and quantify VBNC *C. jejuni* in naturally contaminated manure from experimental pens for up to 72 h. It can be assumed that the transition into VBNC states within manure may pose the challenges encountered in cultivating *Campylobacter* from chicken manure [54]. Indeed, this could also limit the time frame for *Campylobacter* detection in manure by cultivation, as observed in this and previous studies [40]. This might be explained by the different properties manure provides. One driving factor could be the abundance of nutrients, as Yagi et al. [33] found that nutrient-rich conditions induced the VBNC state faster than nutrient-poor conditions. In terms of stability and VBNC formation in soil, a rapid decrease in cultivability and rapid subsequent formation of VBNC *C. jejuni* were found. It is possible that various abiotic factors in soil, such as desiccation, physical entrapment, fluctuating oxygen levels, and biotic factors like competition from soil microflora, may be potential drivers of rapid VBNC formation, which resulted in, on the one hand, faster formation of VBNC *Campylobacter* and, on the other hand, reduced stability compared to water [21].

## 5. Conclusions

This study aimed to investigate the transition and persistence of VBNC *Campylobacter* in the environment and under predefined conditions. Utilizing v-qPCR for the analysis of samples obtained under favorable environmental conditions (rainy, cloudy, moist, low temperature), we were able to display viable *C. jejuni* and, thus, potential VBNC *Campylobacter* in the environment of broiler farms. The results of the experimental trials showed that viable *C. jejuni* cells could be detected in manure for up to 48 h. In our laboratory experiments, we were able to demonstrate that VBNC *Campylobacter* can remain viable over extended periods in different evaluated settings. Specifically, VBNC *Campylobacter* was viable for prolonged periods in water. However, it is important to note that controlled experimental settings do not necessarily reflect and mimic the VBNC *Campylobacter* persistence in nature (the environment). While we found that temperature, desiccation, humidity, and UV light appear to be important environmental drivers for VBNC *Campylobacter*, it should be emphasized, however, that other factors such as strain-specific differences and nutrient availability may also influence the persistence of VBNC *Campylobacter.* It is important to note that PMA alone confirms cell membrane integrity but provides no data on metabolic activity, pathogenicity, or infectivity. To gain deeper insights into VBNC states, further concurrent studies with v-qPCR, colored staining, microscopy, and in vivo/in vitro assays using isolated VBNC *Campylobacter* are of particular interest. To conclude, this comprehensive approach enhances and elucidates the overall understanding of the role of VBNC *Campylobacter* in the poultry reservoir.

## Figures and Tables

**Figure 1 microorganisms-11-02492-f001:**
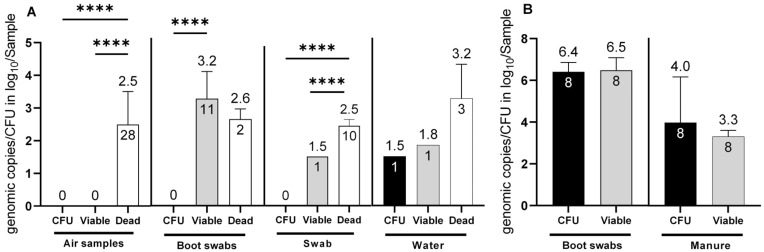
Determination of *C. jejuni* in (**A**) environmental matrices (air samples, boot swabs, swabs and water) and (**B**) barn matrices (boot swabs and manure). Determination of *C. jejuni*: viable cell counts with v-qPCR (log10 viable Cj/sample (air (1 m^3^), boot swabs, gauze swabs, water (50 mL) and manure (5 g)) (grey bars), CFU (log_10_ CFU/Sample) (black bars) and exclusively dead cells (log_10_ total Cj/Sample) (white bars). The error bars depict the standard deviation of the mean counts (shown on top of the bars) (**** *p* < 0.0001). The number of positive samples is indicated within each bar. In total 86 environmental samples (28 air samples, 24 boot swab samples, 18, gauze swabs and 16 water samples from visits 1, 2, 6 and 7) were investigated. Simultaneously, 8 boot swabs and 8 manure samples were investigated from the inside of the barns.

**Figure 2 microorganisms-11-02492-f002:**
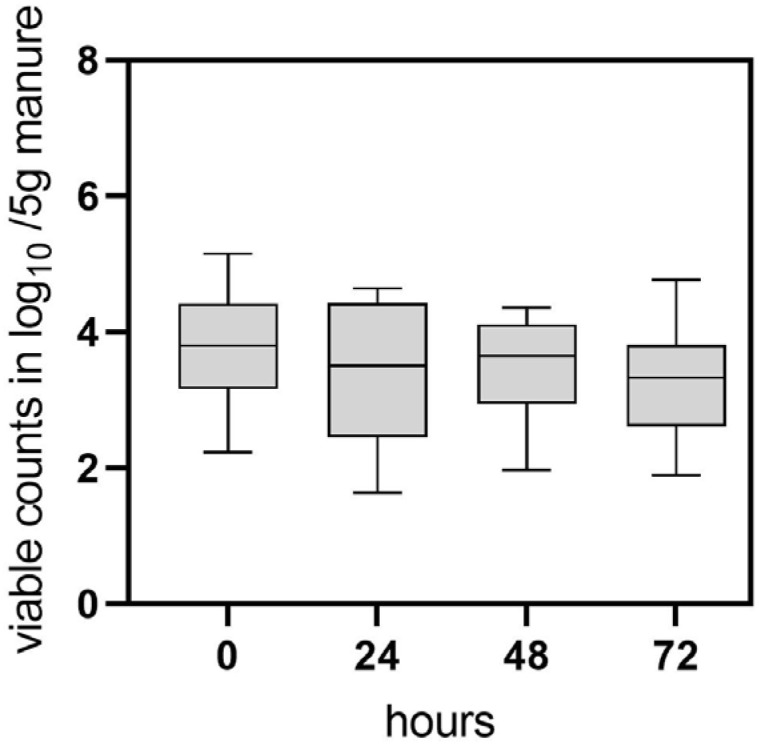
Determination of viable *C. jejuni* BfR-CA-14430 counts from naturally contaminated chicken manure using v-qPCR. At each time point (hours), 16 manure samples were pretreated by homogenization and filtering and investigated. Grey bars depict the mean of positive v-qPCR detections in log_10_ viable Cj/5 g manure at 0 h (*n* = 16), 24 h (*n* = 15) 48 h (*n* = 16), 72 h (*n* = 9). The error bars represent the standard deviation for the mean (grey bar). No CFU was detected at sampling time point 0, while enrichment of *C. jejuni* was possible until 24 h.

**Figure 3 microorganisms-11-02492-f003:**
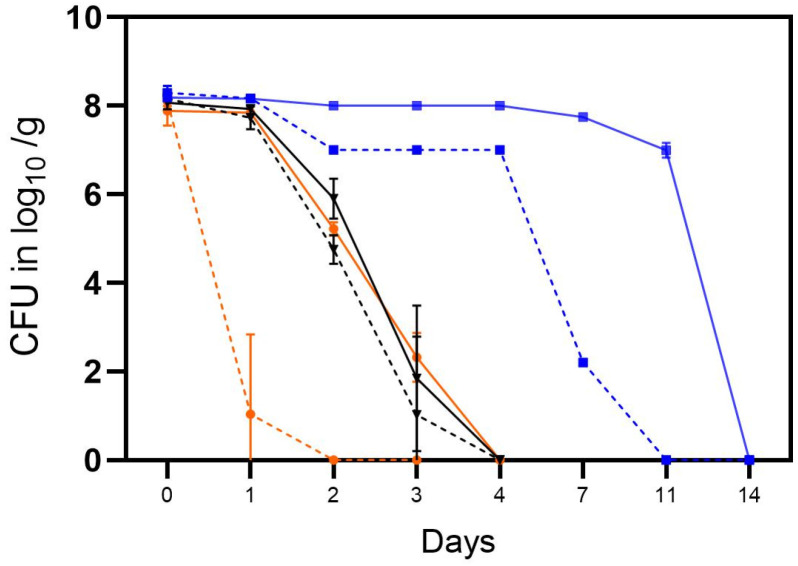
The tenacity of cultivable *C. jejuni* BfR-CA-14443 in the soil in three different habitats: laboratory at RT, 35% RH, daylight (orange), incubator at RT, 9% RH, dark (black), and refrigerator at 4 °C, 64% RH, dark (blue). Counts were determined for closed containers (solid lines) and open containers (dotted lines) and are depicted in log_10_ CFU/g soil. Error bars indicate the standard deviation of the mean counts.

**Figure 4 microorganisms-11-02492-f004:**
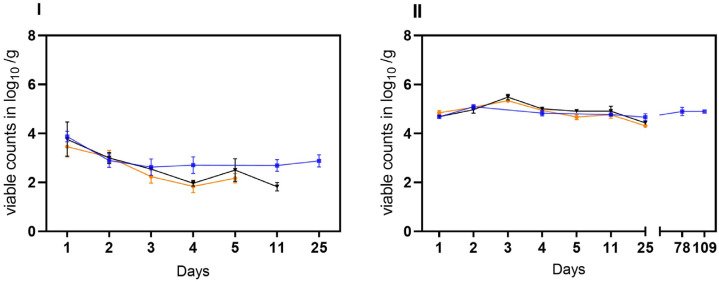
Stability of VBNC *C. jejuni* cells BfR-CA-14443 from raw milk in three habitats: laboratory at 21 °C (orange), incubator at 22 °C (black), and refrigerator at 4 °C (blue). Two different matrices were investigated: soil (**I**) and water (**II**). The sampling frequency is displayed in days. The counts, measured in log_10_ viable Cj/g soil or log_10_ viable Cj/mL water using v-qPCR, are depicted. The error bars indicate the standard deviation for the mean counts.

**Figure 5 microorganisms-11-02492-f005:**
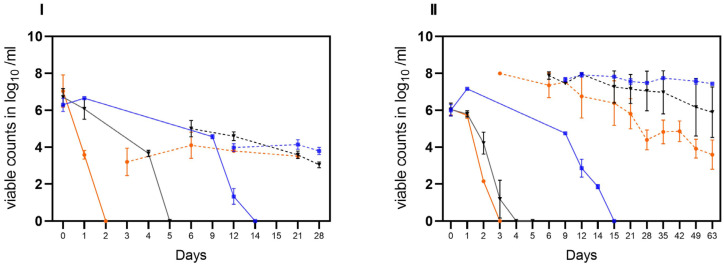
Transition of cultivable *C. jejuni* (BfR-CA-14430) into viable but nonculturable (VBNC) state observed in three different habitats: laboratory at 21 °C (orange), incubator at 22 °C (black), and refrigerator at 4 °C (blue). Two different matrices were investigated: soil (**I**) and water (**II**). The sampling frequency is displayed in days. The counts, measured in log_10_ viable Cj/g soil or log_10_ viable Cj/mL water with v-qPCR (dotted lines) and CFU/g soil or ml water (solid lines) are depicted. The error bars indicate the standard deviation for the mean counts.

**Table 1 microorganisms-11-02492-t001:** Broiler farm visits and weather conditions at the day of sampling, Germany 2019–2020.

Visit	1	2	3	4	5	6	7
Farm	C	A	B	B	C	B	C
Season	Autumn	Winter	Winter	Winter	Winter	Summer	Summer
Weather condition	Cloudy/Overcast	Clear-Fair	Cloudy/Overcast	Overcast/Light rain	Overcast/Light rain	Clear, Overcast/Rain	Overcast/Rain
Average temperature	5.6 °C	0.3 °C	0.9 °C	3.4 °C	4.3 °C	22 °C	19.7
Total precipitation/24 h	0.3 mm	0 mm	0 mm	0.8	8.1 mm	0 mm	0.6 mm
Average humidity percentage	80.4%	83%	79%	96%	85%	71%	77%
Sunlight minutes/24 h	340 m	415 m	78 m	2 m	61 m	554 m	153
*Campylobacter* positive barns	yes	yes	No	No	No	yes	yes

Source: Deutscher Wetterdienst Meteorologisches Observatorium Lindenberg. Viable thermotolerant *Campylobacter* spp. were found in samples collected during visits 1, 2, 6, and 7.

**Table 2 microorganisms-11-02492-t002:** Microhabitats of the laboratory-based trials.

Laboratory (A)	Incubator (B)	Refrigerator (C)
21 ± 0.3 °C	22.2 ± 0.03 °C	4.0 ± 0.1 °C
RH ^(a)^ 35.2 ± 3.5%	RH 98.8 ± 0%	RH 64.3 ± 12%
Sunlight (Windows)	no light	no light

^(a)^ Relative humidity (RH) Incubator B (CB 160; Binder, Tuttlingen, Germany), Refrigerator C (Liebherr GKV 6410, Ochsenhausen, Germany).

**Table 3 microorganisms-11-02492-t003:** Triplex qPCR primers and probes.

Name	5′-3′Sequence	Target	Quelle
Jos-F1 Jos-R1 Jos-P	5′-CCTGCTTAACACAAGTTGAGTAGG-3′ 5′-TTCC TTAGGTACCGTCAGAATTC-3′ 6FAM-TGTCATCCTCCACGCGGCGTTGCTGC-BHQ-1	16s rRNA	Pacholewicz et al., 2019 [27]
Csput-F Csput-R Csput-P	5′-TGGGAAATGTAGCTCTTAATAATATATATC-3′ 5′-CCTTACCAACTA GCTGATACAATATAG-3′ Cy5-CCTCATCCCA TAGCGAAAGCTCTT-BBQ-650	16s rRNA	Pacholewicz et al., 2019 [27]
IPC-ntb2-F IPC-ntb2-R IPC-ntb2-P	5′-ACCACAAT GCCAGAGTGACAAC-3′ 5′-TACCTGGTCTC CAGCTTTCAGTT-3′ HEX-CACGCGCATGAAGTTAGGGGACCA-BHQ-1	rbcMT-T	Anderson et al.,2011 [31]

## Data Availability

Data is contained within the article.

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
