# Peer review of "Detection of Viable but Non-Culturable (VBNC)-Campylobacter in the Environment of Broiler Farms: Innovative Insights Delivered by Propidium Monoazide (PMA)-v-qPCR Analysis"

_microorganisms, 2023, doi:10.3390/microorganisms11102492_

Round 1

Reviewer 1 Report

Lines 1-4: the full names of VBNC and PMA in the title should be provided

Line 16: the full name of PMA should be provided when it was first mentioned

Line 29: the full names of VBNC and PMA in the abstract should be provided

 In the abstract, the risk of Campylobacter to human health and the overall Campylobacter prevalence in broiler farms should be mentioned.

 Line 155: What is the reference for the C. jejuni strain BfR-CA-14430?

 Section 2.5. Determination of VBNC Campylobacter with qPCR: (1) the primers for determining Campylobacter spp. should be provided; (2) the qPCR procedures (temperature, duration, cycles, etc.) need to be mentioned in the text although the related reference is provided; (3) How genomic copies of C. jejuni were calculated? This needs to be provided in the text.

 Some paragraphs in the Methods and Results ((e.g., 2.5. Determination of VBNC Campylobacter with qPCR; 3.1. Field trial)) are too long and they are difficult for readers to follow. They can be improved.

 The statistical results, i.e., P-values need to be mentioned in the text of the Results Section. If P-values are less than 0.05, the significance needs to be reflected in the figures.

 Is the first paper to use PMA dye-v-qPCR for VBNC-Campylobacter detection? How is the technique of PMA dye-v-qPCR applied in livestock production, particularly for pathogen detection? This needs to be briefly introduced in the manuscript (Introduction or Discussion).

Author Response

thank you very much for the time and effort you devoted to our manuscript and the opportunity to submit a revised draft of our manuscript “Detection of VBNC-Campylobacter in the Environment of Broiler Farms: Innovative Insights delivered by PMA-v-qPCR Analysis”. We are very grateful for the comprehensive review and appreciate your helpful suggestions, notes and comments that improved our manuscript. We have carefully revised the manuscript based on your comments. Below, we address your comments point-by-point. Briefly, we have reworked the recommended sections and provided information on the PMA method in the introduction. We have revised some statistical outputs in the text and in our figures. We have attempted to address all comments and implemented them accordingly in the revised version of the manuscript (changes in red).

Sincerely,

Benjamin Reichelt

Lines 1-4: the full names of VBNC and PMA in the title should be provided

Lines 1-4: Thank you for your comment, we added the full names as requested.

Line 16: the full name of PMA should be provided when it was first mentioned

Line 18: Thank you very much for this insightful comment, we added the full names as requested.

Line 29: the full names of VBNC and PMA in the abstract should be provided

Line 31: Former Line 29: Thank you for your comment. We added the full names accordingly.

In the abstract, the risk of Campylobacter to human health and the overall Campylobacter prevalence in broiler farms should be mentioned.

Line 13-14: Thank you for pointing this out. We added relevant information as requested.

 Line 155: What is the reference for the C. jejuni strain BfR-CA-14430?

Line 102: Thank you for pointing this out. The reference for the strain is now provided where it is first introduced.

 Section 2.5. Determination of VBNC Campylobacter with qPCR: (1) the primers for determining Campylobacter spp. should be provided; (2) the qPCR procedures (temperature, duration, cycles, etc.) need to be mentioned in the text although the related reference is provided; (3) How genomic copies of C. jejuni were calculated? This needs to be provided in the text.

Line: 217-243 Thank you for pointing this out, we explained the details of (1-3) in brief as requested.

 Some paragraphs in the Methods and Results ((e.g., 2.5. Determination of VBNC Campylobacter with qPCR;3.1. Field trial)) are too long and they are difficult for readers to follow. They can be improved.

We sincerely thank the reviewer for their valuable suggestions. We have incorporated these suggestions into Section 2.5 (see above). We think section 3.1 contains a lot of data and therefore shortening it further might reduce readability.

 The statistical results, i.e., P-values need to be mentioned in the text of the Results Section. If P-values are less than 0.05, the significance needs to be reflected in the figures.

Lines 253-282: Thank you for that crucial comment. We have added P-values in the text and the corresponding figure as requested.

Is the first paper to use PMA dye-v-qPCR for VBNC-Campylobacter detection? How is the technique of PMA dye-v-qPCR applied in livestock production, particularly for pathogen detection? This needs to be briefly introduced in the manuscript (Introduction or Discussion).

Thank you for your comment. Lines 53 to 64 provide an overview of the existing literature that supports the use of PMA dye-v-qPCR for Campylobacter detection.

Lines 73-75: In response to this suggestion, we have added a brief section on the methodology in the introduction as requested. We believe that this addition enhances the overall comprehensibility of the introduction.

Reviewer 2 Report

This is an excellent well conducted and written study, with an original contribution to the knowledge of detection of VBNC-Campylobacter in the environment of broiler farms: Innovative Insights delivered by PMA-v-qPCR 3 analysis.

I encourage it’s acceptance after appropriate minor modifications as outlined below:

L48: The documentation about the public health importance and the presence of Campylobacter spp. in the food-chain over the last decade must be improved with some recently published valuable articles in Antibiotics (e.g. https://doi.org/10.3390/antibiotics11121713 and https://doi.org/10.1016/j.cmi.2015.11.019). These manuscript should be consulted and cited.

L64,66: “we analyzed naturally contaminated chicken manure obtained from experimental pens. To further expand our under standing, we investigated” – Please rephrase. The personal writing style (we, our etc.) is not allowed in scientific style (scientific manuscripts). Please revise this aspect and use impersonal style in the main manuscript.

L80: I’m wondering if the farms established some biosecurity measures to combat the wildlife and the potential harm and pathogens that wild species can spread in farm areas.

L115: “cooling box” – Please mention the transport temperature.

L521: Within the conclusion section, I would like to advise the authors to honestly underline the study limitations (if they exist), and mention further perspectives in the studied research area.

Author Response

thank you very much for your crucial comments and the opportunity to submit a revised draft of our manuscript “Detection of VBNC-Campylobacter in the Environment of Broiler Farms: Innovative Insights delivered by PMA-v-qPCR Analysis”. We appreciate the time and effort that you dedicated to providing feedback on our manuscript and are grateful for the insightful comments on our paper. We have made adjustments to incorporate impersonal style in the complete manuscript. We added documentation about the significance of public health importance as requested. Following, we reply point-by-point to your comments. We have attempted to address all comments and implemented them accordingly in the revised version of the manuscript (changes in red).

Sincerely,

Benjamin Reichelt

 L48: The documentation about the public health importance and the presence of Campylobacter spp. in the food-chain over the last decade must be improved with some recently published valuable articles in Antibiotics (e.g. https://doi.org/10.3390/antibiotics11121713 and https://doi.org/10.1016/j.cmi.2015.11.019). These manuscript should be consulted and cited.

Line 38 – 43: Thank you for pointing this out. We incorporated the information and cited the papers in the introduction as recommended.

L64,66: “we analyzed naturally contaminated chicken manure obtained from experimental pens. To further expand our understanding, we investigated” – Please rephrase. The personal writing style (we, our etc.) is not allowed in scientific style (scientific manuscripts). Please revise this aspect and use impersonal style in the main manuscript.

We would like to express our gratitude to the reviewer for this valuable feedback. In response to these suggestions, we have rephrased all content in the main manuscript accordingly. Personal writing style is now reserved solely for the Abstract and Conclusion sections. If the reviewer advises otherwise, we will make changes accordingly.

L80: I’m wondering if the farms established some biosecurity measures to combat the wildlife and the potential harm and pathogens that wild species can spread in farm areas.

Thank you for your insightful comment. We have confidence in farmers adopting best practices in line with established DVG guidelines, and they are also obligated to follow the prescribed pest monitoring and control protocols as part of their participation in the QS Agriculture Animal Husbandry program.

L115: “cooling box” – Please mention the transport temperature.

Line 126-127: Thank you for your note. We included the temperature (~4 C°)

L521: Within the conclusion section, I would like to advise the authors to honestly underline the study limitations (if they exist), and mention further perspectives in the studied research area.

Line 581 – 586 We thank the reviewer for this suggestion. As requested, we included limitations and further perspectives in the conclusion section.